# Real-time PCR assay for detection and differentiation of *Coccidioides immitis* and *Coccidioides posadasii* from culture and clinical specimens

Sudha Chaturvedi[1,2]*, Tanya R. Victor[1], Anuradha Marathe[1], Ketevan Sidamonidze[1¤], Kelly L. Crucillo[3], Vishnu Chaturvedi[1]*

1 Mycology Laboratory, Wadsworth Center, New York State Department of Health, Albany, New York, United States of America, 2 Department of Biomedical Sciences, University at Albany, Albany, New York, United States of America, 3 Coccidioidomycosis Serology Laboratory, Department of Medical Microbiology and Immunology, University of California School of Medicine, Davis, California, United States of America

¤ Current address: Lugar Center for Public Health Research, 0184 Tbilisi, Georgia.
* Sudha.Chaturvedi@health.ny.gov (SC); Vishnu.Chaturvedi@health.ny.gov (VC)

**Data Availability Statement:** All relevant data are within the manuscript and its Supporting Information files.

## Abstract

Coccidioidomycosis (Valley fever) is a pulmonary and systemic fungal disease with increasing incidence and expanding endemic areas. The differentiation of etiologic agents *Coccidioides immitis* and *C. posadasii* remains problematic in the clinical laboratories as conventional PCR and satellite typing schemes are not facile. Therefore, we developed Cy5- and FAM-labeled TaqMan-probes for duplex real-time PCR assay for rapid differentiation of *C. immitis* and *C. posadasii* from culture and clinical specimens. The *RRA2* gene encoding proline-rich antigen 2, specific for *Coccidioides* genus, was the source for the first set of primers and probe. *Coccidioides immitis* contig 2.2 (GenBank: AAEC02000002.1) was used to design the second set of primers and probe. The second primers/probe did not amplify the corresponding *C. posadasii* DNA, because of an 86-bp deletion in the contig. The assay was highly sensitive with limit of detection of 0.1 pg gDNA/PCR reaction, which was equivalent to approximately ten genome copies of *C. immitis* or *C. posadasii*. The assay was highly specific with no cross-reactivity to the wide range of fungal and bacterial pathogens. Retrospective analysis of fungal isolates and primary specimens submitted from 1995 to 2020 confirmed 168 isolates and four primary specimens as *C. posadasii* and 30 isolates as *C. immitis* from human coccidioidomycosis cases, while all eight primary samples from two animals (rhesus monkey and rhinoceros) were confirmed as *C. posadasii*. A preliminary analysis of cerebrospinal fluid (CSF) and pleural fluid samples showed positive correlation between serology tests and real-time PCR for two of the 15 samples. The *Coccidioides* spp. duplex real-time PCR will allow rapid differentiation of *C. immitis* and *C. posadasii* from clinical specimens and further augment the treatment and surveillance of coccidioidomycosis.

**Funding:** This work was supported partly by funds from the Wadsworth Center (Sc & Vc), the New York State Department of Health (NYSDOH), and the Centers for the Disease Control and Prevention (CDC) grant number NU50CK000516 (Sc &Vc). The funders played no role in the study design, data collection and analysis, decision to publish, or preparation of the manuscript.

**Competing interests:** The authors have declared that no competing interests exist.

## Author summary

Coccidioidomycosis (Valley fever) is a fungal disease caused by two closely related pathogens: *Coccidioides immitis* and *C. posadasii*. The numbers of Valley fever cases in the US show an upward trend, and possibly many more cases go unrecognized annually. Rapid and accurate diagnostic tests are needed for the differentiation of *C. immitis* and *C. posadasii*. We developed and validated a TaqMan real-time PCR assay that works well for both isolates and primary specimens. The new laboratory test for *C. immitis* and *C. posadasii* will likely improve both treatment management and surveillance of coccidioidomycosis.

## Introduction

Coccidioidomycosis (Valley fever) is a fungal disease caused by two closely related pathogens: *Coccidioides immitis* and *C. posadasii*. *Coccidioides immitis* is endemic to the San Joaquin Valley of California. *Coccidioides posadasii* is found in the desert regions of the southwestern United States, including Arizona, Utah, New Mexico, and West Texas, and parts of Mexico, Argentina, Paraguay, and Central America [1]. *Coccidioides immitis* is likely to present outside the recognized endemic area as evident from human infections and DNA-positive soil samples from Washington state [2,3].

The number of Valley fever cases has increased over the years with 2,271 cases reported in 1998, and 18, 407 cases in 2019 (https://www.cdc.gov/fungal/diseases/coccidioidomycosis/statistics.html). Notably, in 2011, more than 20,000 cases were reported in the US, twice as many cases as tuberculosis [4]. Further documentation of yearly Valley fever cases from Arizona, California, Nevada, New Mexico, Utah, and other jurisdictions are published in the recent updates (https://www.cdc.gov/fungal/diseases/coccidioidomycosis/statistics.html) [5,6]. It is believed that *C. immitis* and *C. posadasii* infect 'tens of thousands' of people per year and many of whom are sick without knowing the cause or have cases so mild that they are not detected [4]. The rise in *Coccidioides* infection is attributed to increased travel, relocation to endemic areas, and possible broader distribution of *C. immitis* and *C. posadasii* than previously recognized [7,8]. Arthroconidia produced by these fungi are highly infectious, and climate change, including dry and hot weather followed by dust storms makes these conidia easily air borne to cause pulmonary infection [9]. Lung infections typically resolve rapidly leaving the patient with a robust acquired immunity to re-infection [10]. However, in some individuals, the disease may progress to a chronic pulmonary condition or to a systemic disease involving the meninges, bones, joints, subcutaneous, and cutaneous tissues [11]. Identifying *Coccidioides* to the species level is likely beneficial in the proper treatment of the patients and disease surveillance. Such a laboratory test could also become an important tool in ongoing studies aimed at defining similarities and differences in Valley fever caused by *C. immitis* and *C. posadasii* [7,8,12].

*Coccidioides immitis* and *C. posadasii* are not easily differentiated in the clinical laboratory on account of similar morphology. *Coccidioides posadasii* was first differentiated from *C. immitis* by microsatellite analysis [13,14]. However, this approach was time consuming and technically challenging. Subsequently, several real-time PCR assays, based on LightCycler and TaqMan chemistries, were developed to differentiate *C. immitis* from *C. posadasii* [15,16]. Both approaches rely on a single nucleotide polymorphism in different regions of the target gene, and it may have limitations when large number of strains of *C. immitis* and *C. posadasii* are tested. Umeyama et al [17] described disparity PCR using species-specific primers designed from the Ci45815 PCR fragment (GenBank AB597180.1). Contiguous deletion of 86-bp

nucleotides in corresponding *C. posadasii* contig Cp45810 PCR fragment (GenBank AB597183.1) resulted in convenient distinction of *C. immitis* from *C. posadasii* by conventional PCR. In the present study, we describe a TaqMan duplex real-time PCR assay using *PRA2* gene encoding proline-rich antigen specific for the *Coccidioides* genus and *C. immitis* PCR fragment Ci45815 specific for *C. immitis*. The first set of primers and probe targets the *PRA2* gene specific for *Coccidioides* genus. The second set of primers and probe targets *C. immitis* contig Ci45815 (CiC). The duplex real-time PCR assay was compared against redesigned diagnostic conventional PCR with 100% concordance for culture isolates. We show that duplex real-time PCR assay is highly sensitive (10 gene copies) and identifies *C. immitis* and *C. posadasii* from culture and primary specimens. The reliable, and sensitive duplex real-time PCR assay will allow rapid differentiation of *C. immitis* and *C. posadasii* from culture and clinical specimens, and further augment the treatment and surveillance of coccidioidomycosis.

## Materials and methods

### *Coccidioides* isolates and primary specimens

Two hundred and seventeen suspected isolates of *Coccidioides* species, 10 formalin-fixed paraffin-embedded tissues, three vials of rhesus monkey kidney (RhMK) cell (Lot # A491216B) [18], six lung tissue, three cerebrospinal fluid (CSF), two bone marrow, one plural fluid, and one whole blood sample, received as a part of reference service from 1995- to 2020, from various diagnostic laboratories of the New York State and neighboring states' laboratories were part of this investigation. Seventeen isolates of *C. immitis* were received from the Microbial Diseases Laboratory, California State Department of Public Health. All cultures were stored as sterile water and glycerol stocks at 30°C and -80°C, respectively. Fifteen residual CSF and pleural fluids, which tested positive for coccidioidomycosis by immunodiffusion and complement fixation at the University of California Davis, were also included in the study.

### DNA extraction

Extraction of DNA from culture of *Coccidioides* spp. was carried out in the biological safety cabinet 2 (BSC 2) in the biological safety level 3 (BSL 3) laboratory, and DNA extraction from primary human and animal specimens was carried out in BSC 2 in the BSL 2 laboratory. Qiagen DNA mini kit on the QIAcube automated extractor was used for DNA extraction from all sample types. In brief, for *Coccidioides* isolates, approximately 5 x 5 mm size of fungal mat grown on SDA slant for 7 to 10 days was removed by sterile loop and suspended in lysis buffer containing approximately 0.2 g of glass beads and incubated for 1 h at 90°C in the BSL 3 laboratory. Heat-killed fungal cells along with the beads was brought to the BSL 2 laboratory, bead-beating in the Precellys homogenizer at 6,500 rpm for 15 sec. for three times using program number 5 (6500-3x60-015). The homogenized fungal suspension was transferred to a 2-ml screw-cap tube, leaving behind the beads. DNA from homogenized samples was extracted using the Qiagen DNA mini kit in the QIAcube semiautomated DNA extractor, resulting in 50 µl of eluted DNA. For DNA extraction from fungal mat grown in the RhMK cell line, approximately 2 ml RhMK medium containing fungal growth was first centrifuged at 12,000 RPM; the supernatant was decanted, and fungal pellet was suspended in lysis buffer containing beads and heat-killed at 90°C for 1 h in the BSL 3 laboratory followed by DNA extraction in the BSL 2 laboratory as described above. Formalin-fixed paraffin-embedded tissue sections were processed by placing 10 sections of 10 µm each into 2-ml tube containing 1 ml of xylene (Sigma Aldrich). The sections were then incubated in xylene for 5 min at room temperature and xylene was removed with fine-tip disposable pipette. A second incubation of 1 ml of xylene was performed followed by centrifugation and removal of xylene. To the sample pallet, 1 ml of

100% ethanol was added, gently vortexed and incubated at room temperature for 5 min, followed by centrifugation and removal of ethanol. The pellet was washed one more time with 1 ml of 100% ethanol. The tissue was dried at room temperature for 15–20 min, followed by addition of lysis buffer, incubation at 70˚C for 1 h, bead-beating and DNA extraction using QIAcube as described above. For other primary samples (blood, CSF, pleural fluid, and bone marrow), approximately 400 μl of each sample was directly mixed with lysis buffer containing beads, heated at 56˚C for 10 min followed by bead beating and DNA extraction in QIAcube as described above. Lung tissues were minced with sterile blade followed by lysis and DNA extraction as described for other primary samples. All extracted DNAs were stored at −80˚C. DNA of fungi (yeasts and molds) other than *Coccidioides* spp. were procured from the Wadsworth Center Mycology Laboratory (WCML) DNA Collection Repository. DNA from bacterial species including *Bacillus megaterium*, *Escherichia coli*, *Nocardia farcinica*, *Pseudomonas aeruginosa*, and *Streptococcus pneumonia* was procured from the Wadsworth Center Bacteriology Laboratory.

## Modified conventional PCR

A modification of conventional diagnostic PCR described by Umeyama et. al [17] was used. The primers were designed from the conserved region of Ci45815 (GenBank No. AB597180.1) and Cp45810 (GenBank No. AB597183.1) flanking the deleted region of *C. posadasii* (S1A Fig). The nucleotide sequences of the diagnostic primers were V2119 5'-CCGGGTACTCCG TACATCAC-3', and V2120 5'-ATGCGTGAAGCCAATTCTTT-3' and the PCR conditions were initial denaturation at 95˚C for 1 min followed by 30 cycles consisted of denaturation at 94˚C for 1 min, annealing at 55˚C for 1 min, and extension at 68˚C for 1 min followed by final extension at 68˚C for 3 min.

## Duplex real-time PCR assay

The *PRA2* gene and contig Ci45815/Cp45810 [17] were used to design the duplex real-time PCR assay (S1B and S1C Fig). The nucleotide sequences of primers and probe for *PRA2* gene target were forward primer V1753 5'GTGCGAGAAGTTGACCGACTT-3', reverse primer V1754 5'AGGCGTGATCTTTCCTGGAA-3', probe V1755 5'-Cy5'-AAGTGCCACTGCGC CAAGCCC-3BHQ and primers and probe for contig Ci45815 target were forward primer V2116 5'-GGTGAAATGCCCGAAAAGAG-3' reverse primers V2118 5'-CCAATCCTTAGG TAACCGTGAG-3', and probe V2117 5'/56FAMTTGCACTTT/ZEN/CGTTGACTAGCCGC/ 3IABkFQ/-3'. Reaction for each real-time PCR contained 1× PerfeCTa multiplex qPCR ToughMix (Quanta Biosciences), a 1,000 nM concentration of primers and a 250 nM concentration of probes, and 2 μl of genomic DNA (approximately 1 to 10 ng) from isolates or 5 μl of tissue DNA extracted from primary clinical specimens in a final volume of 20 μl. Each PCR run also included 2 μl (1 ng) of positive extraction control (C735; *C. posadasii*), 2 μl (1 ng) of positive amplification controls (C735, *C. posadasii* and 249, *C. immitis*), and 2 μl of negative extraction (extraction reagents only) and negative amplification (sterilized nuclease-free water) controls. Parallel to each PCR assay, inhibitory PCR was also performed by incorporating 1 ng of *Coccidioides* gDNA into each primary clinical DNA sample. The unidirectional workflow kept the reagent preparation, specimen preparation, and amplification and detection areas separate to avoid cross-contamination. Cycling conditions on the ABI 7500 FAST system (Applied Biosystems, Thermo Fisher Scientific Inc., Waltham, MA) were initial denaturation at 95˚C for 20 s, followed by 45 cycles of 95˚C for 3 s and 60˚C for 30 s. Based on the limit of detection (LOD), a cycle threshold (*Ct*) value of ≤38 was reported as positive; and >38 was

reported as negative. For all primary samples, specimens were reported as inconclusive if PCR inhibition was observed for the primary specimens.

### Analytical sensitivity, specificity, and reproducibility of the duplex real-time PCR assay

The 10-fold serial dilutions of genomic DNA from one isolate each of *C. posadasii* (C-735) and *C. immits* (249) were used to assess the analytical sensitivity of the duplex real-time PCR assay. The assay specificity was assessed by using an extensive DNA panel comprising various fungi both closely and distantly related to *Coccidioides* spp. and few bacterial pathogens primarily responsible for causing pulmonary infection (S1 Table). The assay reproducibility was determined by using varying concentration of gDNA from three isolates each of *C. posadasii* and *C. immitis*, ran on three different days in duplicate (inter-assay reproducibility), and on the same day in triplicate (intra-assay reproducibility). The assay's precision was assessed using blinded panel with varying concentration of gDNA from 20 isolates each of *C. posadassii* and *C. immitis*, and 10 isolates of fungi other than *Coccidioides* spp.

### Statistical analysis

The GraphPad Prism 8.0 software for macOS was used for calculation of mean, standard deviation, and percent coefficient of variance (CV) for Ct values generated for assay sensitivity and reproducibility. The student t test was used for the analysis of means, and variance, and *P* value of < 0.05 was considered as statistically significant.

## Results

### Modified conventional PCR assay

Initially, we modified conventional diagnostic PCR assay using species-specific primers designed from the Ci45815 PCR fragment (GenBank AB597180.1) described by Umeyama et al [17]. The modification reduced the amplicon length down to 200-bp for *C. immitis* and 114-bp for *C. posadasii*, yielding better electrophoretic separation (Fig 1). Also, PCR efficiency was higher, with a smaller amplicon size. Of 234 human isolates suspected of *Coccidioides* were investigated, 172 were *C. posadasii*, and 43 were *C. immitis* by conventional PCR, and 19 isolates were negative for *Coccidioides* spp. (Table 1). This approach was not successful for the analysis of *Coccidioides* DNA from primary specimens except for three RhMK cell samples (Tables 1 and S2). Overall, our results showed that the conventional diagnostic PCR could be used successfully to identify *C. immitis* and *C. posadasii* from culture isolates but not from the primary specimens.

### Duplex real-time PCR assay sensitivity, specificity, and reproducibility

We designed a duplex real-time PCR assay with the first set of primers and probe targeting *PRA2* gene identifying *Coccidioides* species (S1B Fig), and the second set of primers and probe targeting *C. immitis* contig Ci45815 (CiC) identifying *C. immitis* but not *C. posadasii* due to 86-bp deletion in the corresponding *C. posadasii* contig 45810 (S1C Fig). The duplex real-time PCR assay was highly sensitive, with the limit of detection was 0.1 pg gDNA/PCR reaction, which was equivalent to approximately ten genome copies of *C. immitis* or *C. posadasii* (Fig 2 and Table 2). None of the other fungal or bacterial DNA yielded any Ct value, confirming the high specificity of the duplex real-time PCR assay (S1 Table). The assay was highly reproducible as it yielded correct ID for *C. immitis* and *C. posadasii* when varying concentration of gDNA tested within the same day or on different days (S3A and S3B Table), and when samples

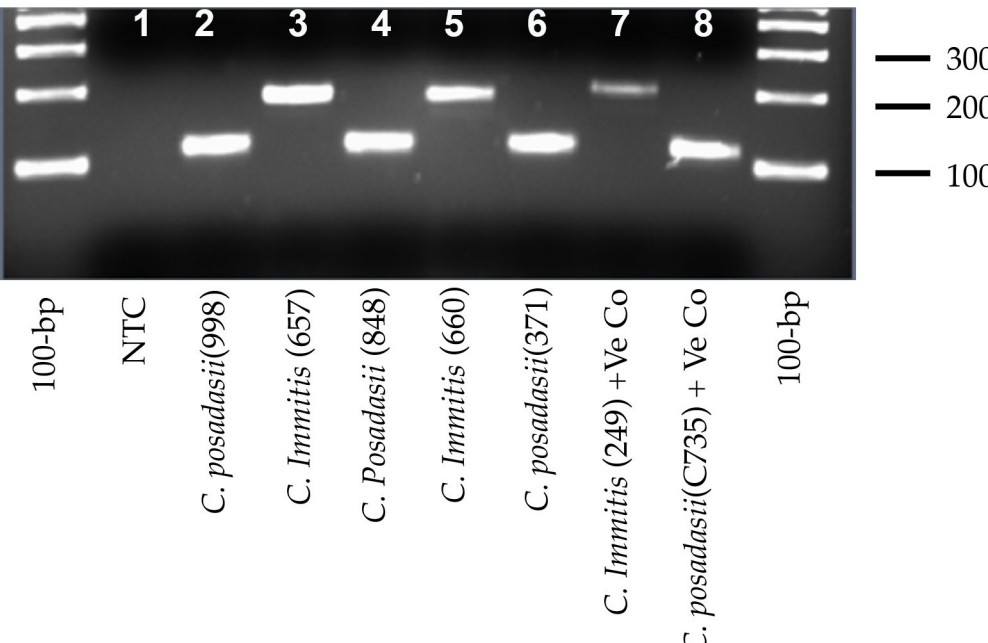

**Fig 1. A modified conventional diagnostic PCR assay.** A modification of assay described by Umeyama et al. (2006) allowed generation of smaller amplicons and better electrophoretic separation while maintaining test accuracy. First and last lane 100-bp DNA ladder; lane 1 non-template control (NTC), lanes 2, 4, 6, and 8 *C. posadasii*; lanes 3, 5, and 7 *C. immitis*.

**Table 1. Results of histopathology, serology, culture, conventional and duplex real-time.**

| Sample | Type | No | Histopathology Positive/Negative | Serology Positive/ Negative (ND) | Culture ID$ Positive/ Negative | Conventional PCR | | Duplex PCR | | Sensitivity (%) | Specificity (%) | PPV | NPV |
|---|---|---|---|---|---|---|---|---|---|---|---|---|---|
| | | | | | | 114-bp | 200-bp | *Cp* (Cy5 +/FAM-) | *Ci* (CY5 +/FAM+) | | | | |
| Primary specimens | Paraffin Tissue | 10 | 6/4 | NA | NA | 0 | 0 | 6+3[#] | 0 | 100 | NA | 100 | NA |
| | RhMK Cell | 3 | NA | NA | 3/0 | 3 | 0 | 3 | 0 | 100 | NA | 100 | NA |
| | CSF | 9 | NA | 6/0 (2) | 0 | 0 | 0 | 0 | 0 | NA | NA | NA | NA |
| | Pleural Fluid | 10 | NA | 3/6 (1) | 0 | 0 | 0 | 2 | 0 | 100 | 86 | 67 | 100 |
| | Tissues | 10 | ND | NA | 0 | 0 | 0 | 0 | 0 | 0 | 0 | 0 | 0 |
| Culture | California* | 17 | NA | NA | 17/0 | 4 | 13 | 4 | 13 | 100 | NA | 100 | NA |
| | New York@ | 217 | NA | NA | 198/19 | 168 | 30 | 168 | 30 | 100 | 100 | 100 | 100 |

NA = Not Applicable; ND = Not Done

The histopathology or serology or culture was used for different specimen types to determine the performance of *Coccidioides* duplex real-time PCR assay.

$Culture was identified either by GenProbe (Hologic Inc.), or sequencing of the ribosomal gene (ITS), or by *Coccidioides immitis*/*C. posadasii*) lab-developed, and Clinical Laboratory Evaluation Program-NYSDOH-approved single plex real-time PCR assay.

#Histopathology was not done for four samples of which three were positive by *Coccidioides* duplex real-time PCR. Therefore, they were removed from the calculation for PCR accuracy.

* Received as *C. immitis*

@Suspected isolates of *Coccidioides* spp. received as part of reference testing

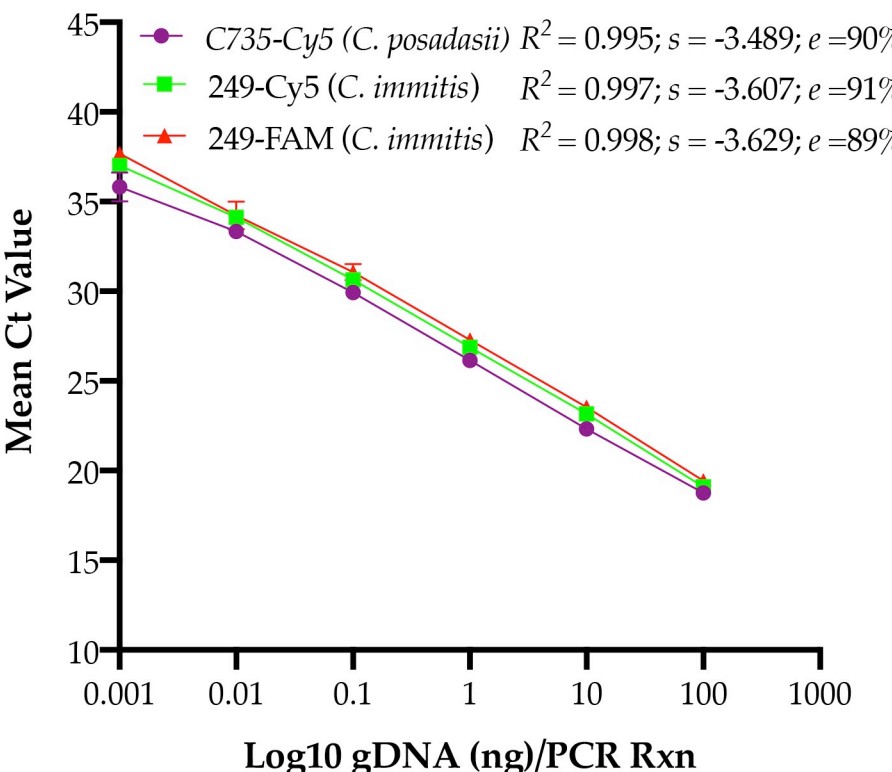

**Fig 2. *Coccidioides* duplex real-time PCR assay sensitivity.** Genomic DNA from two control strains of *Coccidioides* species, *C. immitis* (249) and *C. posadasii* (C-735) were serially diluted and tested in triplicate in duplex real-time PCR assay. The assay was linear over 5 orders of magnitude and the limit of detection was 0.001 ng or 1 picogram gDNA/PCR reactions at 45 PCR cycles confirming high sensitivity. As indicated, the assay targeting *CiC* gene did not yield Ct values against *C. posadasii* DNA. The test was repeated with similar results.

**Table 2. *Coccidioides* duplex real-time PCR assay sensitivity.**

| gDNA ng/PCRRxn | Probe | *C. posadasii* (C-735) | | | Mean Ct ± SD | %CV | *C. immitis* (249) | | | Mean Ct ± SD | % CV |
|---|---|---|---|---|---|---|---|---|---|---|---|
| | | Ct 1 | Ct 2 | Ct 3 | | | Ct 1 | Ct 2 | Ct3 | | |
| 100 | *PRA2 (Cy5)* | 18.66 | 18.86 | 18.7 | 18.74 ± 0.106 | 0.565 | 19.04 | 19.14 | 19.20 | 19.12 ± 0.08 | 0.423 |
| 10 | | 22.36 | 22.28 | 22.25 | 22.29 ± 0.057 | 0.255 | 23.16 | 23.58 | 23.18 | 23.16 ± 0.012 | 0.050 |
| 1 | | 26.34 | 25.96 | 26.56 | 26.28 ± 0.304 | 1.155 | 27.00 | 27.17 | 26.80 | 26.85 ± 0.129 | 0.479 |
| 0.1 | | 29.80 | 30.05 | 29.89 | 29.91 ± 0.127 | 0.423 | 30.83 | 30.75 | 30.60 | 30.62 ± 0.196 | 0.640 |
| 0.01 | | 33.21 | 33.47 | 33.3 | 33.32 ± 0.132 | 0.396 | 33.97 | 33.69 | 34.51 | 34.24 ± 0.266 | 0.776 |
| 0.001 | | 36.38 | 35.25 | 35.3 | 35.64 ± 0.638 | 1.791 | 37.37 | 36.65 | 37.32 | 37.10 ± 0.397 | 1.070 |
| 0.0001 | | 0.0 | 0.0 | 0.0 | | | 0.0 | 0.0 | 0.0 | 0.0 | |
| 100 | *CiC (FAM)* | 0.0 | 0.0 | 0.0 | | | 19.40 | 19.47 | 19.52 | 19.45 ± 0.051 | 0.254 |
| 10 | | 0.0 | 0.0 | 0.0 | | | 23.58 | 23.50 | 23.55 | 23.54 ± 0.040 | 0.172 |
| 1 | | 0.0 | 0.0 | 0.0 | | | 27.17 | 27.40 | 27.35 | 27.30 ± 0.121 | 0.443 |
| 0.1 | | 0.0 | 0.0 | 0.0 | | | 30.75 | 31.40 | 31.00 | 31.05 ± 0.328 | 1.056 |
| 0.01 | | 0.0 | 0.0 | 0.0 | | | 33.69 | 34.78 | 34.50 | 34.32 ± 0.566 | 1.649 |
| 0.001 | | 0.0 | 0.0 | 0.0 | | | 37.83 | 37.50 | 37.60 | 37.64 ± 0.169 | 0.450 |
| 0.0001 | | 0.0 | 0.0 | 0.0 | | | 0.0 | 0.0 | 0.0 | 0.0 | |

**Table 3. Test interpretation of *Coccidioides* duplex real-time PCR assay.**

| Test Interpretation | Probe | |
|---|---|---|
| | ***Prp2* (Cy5)** | ***CiC* (FAM)** |
| *Coccidioides immitis* | Positive | Positive |
| *Coccidioides posadasii* | Positive | Negative |
| Other Fungi | Negative | Negative |
| Bacteria | Negative | Negative |

were blinded by one operator and tested by another operator (S4 Table). Since *Coccidioides* genus comprises only two species, positive results with both probes identified *C. immitis* while the positive, negative results with *PRA2* and Ci45815 contig (CiC) probes, respectively, identified *C. posadasii* (Table 3).

### Performance of the duplex real-time PCR assay

The duplex real-time PCR assay correctly identified all cultures of *Coccidioides* species to either *C. immitis* or *C. posadasii*, and the results were corroborated with modified diagnostic PCR assay, confirming the high utility of the duplex real-time PCR assay (Table 1). Of 217 human fungal isolates suspected of *Coccidioides* received for reference testing, 168 were identified as *C. posadasii*, representing 125 cases, and 30 isolates were identified as *C. immitis*, representing 23 cases (Fig 3). Of 17 isolates received from California, 13 were confirmed as *C. immtits* and four were confirmed as *C. posadasii* (Tables 1 and S5). Next, we assessed the duplex real-time PCR assay's ability to detect *Coccidioides* species DNA from 42 primary specimens. Nine of the

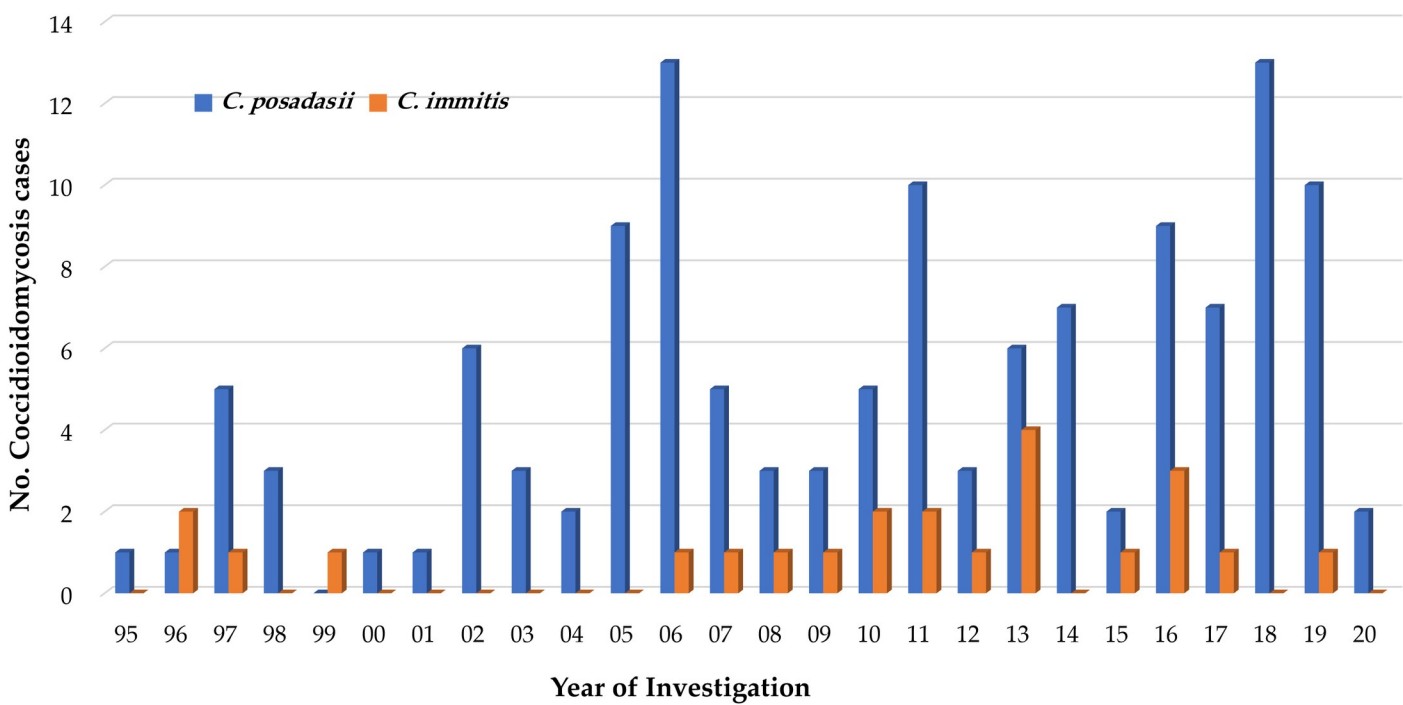

**Fig 3. Human coccidioidomycosis cases from 1995 to 2020 as part of reference diagnostic service.** Isolates received from various facilities from New York and neighboring states were analyzed retrospectively by newly developed duplex real-time PCR assay. Of 152 cases, 129 were confirmed as being caused by *C. posadasii*, and 23 were caused by *C. immitis*. The predominance of *C. posadasii* over *C. immitis* was observed in the isolates analyzed. Four of the 129 cases were defined based on primary specimens positive for *C. posadasii* DNA.

ten paraffin-embedded tissues including four human and five animal, and three vials of RhMK cells were positive for *C. posadassi* DNA (S2 Table). All of the three RhMK cell vials positive for C. *posadasii* DNA also yielded *Coccidioides* in culture, and subsequently confirmed as *C. posadasii* (Tables 1 and S2). We also evaluated 15 residual CSF and pleural fluid samples, which were earlier tested positive for coccidoidomycosis by immunodiffusion and complement fixations tests. Of these samples, only two were positive for *C. posadasii* DNA by duplex real-time PCR assay (Tables 1 and S6).

## Discussion

PCR-based fungal diagnostic methods offer accuracy, sensitivity, and speed of identification, and the use of DNA instead of highly infectious live cultures. Microsatellite analysis has been the primary molecular method used to distinguish the two species of *Coccidioides* [13,14]. However, this method is both time consuming and technically challenging for clinical and reference laboratories. The light-cycler PCR assay developed by Binnicker et al. [19] was highly sensitive and specific for *Coccidioides* species, but this assay could not distinguish *C. immitis* from *C. posadasii*. Umeyama et al. [17] described disparity PCR using species-specific primers that resulted in different size amplicons for a convenient distinction of *C. immitis* from *C. posadasii* by conventional PCR. In the present study, we modified the Umeyama et. al. [17] disparity PCR, which led to the reduction in amplicon size of 200-bp for *C. immitis* and 114-bp for *C. posadasii*. The conventional diagnostic PCR assay was excellent for the identification of *C. immitis* and *C. posadasii* from culture, but it was not good for the analysis of *Coccidioides* DNA from primary specimens. One reason could be that the amount of *Coccidioides* DNA present in the primary samples were below the threshold for detection by conventional PCR, or the template was degraded beyond the capacity of the conventional PCR to yield a positive result. PCR allelic-discrimination assay, referred to as CocciDiff, was developed using TaqMan chemistry [16]. This assay could effectively identify *Coccidioides* cultures and could accurately distinguish *C. immitis* from *C. posadasii* based upon the presence of an individual highly informative canonical SNP [16]. CocciDiff has yet to be validated with clinical specimens. Although, the CocciDiff assay appears to be quite promising, the assay specificity based on canonical SNP can be problematic if such mutations are not consistently present in a given strain of *Coccidioides*. In the present study, while developing a duplex real-time PCR assay, we used more stringent conditions where the first set of primers and probe targeting *PRA2* gene identified *Coccidioides* to the genus level, and the second set of primers and probe targeting *C. immitis* contig, Ci45815 (CiC), identified *C. immitis* but not *C. posadasii* due to a deletion of 86-bp sequence in the corresponding Cp45810 contig. Since there are only two species within the *Coccidioides* genus, the positive results with both probes of the duplex assay allowed the identification of *C. immitis* while positive, negative results with probes *PRA2* and Ci45815, respectively, led to the identification of *C. posadasii*.

Of 168 *C. posadasii*, 151 isolates were from patients from New York, followed by eight isolates from patients from New Jersey, two isolates each from patients from Pennsylvania and Arizona, one isolate from a patient from Washington DC. All 30 *C. immitis* isolates were from patients from NY. Since *Coccidioides* is not found in the soil of New York and other states in the northeast region, the findings of positive isolates suggested that the patients acquired *Coccidioides* infection while travelling to a known area of endemicity. These findings are consistent with our earlier study where travel history to endemic areas for *Coccidioides* was linked to positive clinical cases from New York [20]. In the present study, we confirmed over five-fold more isolates as *C. posadasii* than as *C. immitis*; this observation supports a previous publication on possible larger population size and more diverse distribution of *C. posadasii* vis-a-vis

*C. immitis* [21]. Among a smaller sampling of isolates from California, we found four *C. posadasii* isolates. Also, positive RhMK cell samples from a monkey housed in California were also identified as *C. posadasii*. These results raised the possibility that a small niche of *C. posadasii* exists in California or patients might have traveled to areas where *C. posadasii* is endemic. Further studies are needed to delineate the geographic distribution of *C. immitis* and *C. posadasii* and the areas of overlap if any.

Serology tests are the mainstay of coccidioidomycosis diagnosis [22,23]. Our study included a small sample size of serology-positive clinical specimens, but the findings do not permit firm correlation with real-time PCR test. The gold standard for the diagnosis of coccidioidomycosis is the culture of the organism from primary specimens. Culture is highly sensitive, and the DNA probe for confirmatory testing of culture isolates has yielded excellent specificity [24,25]. However, growth in culture may take several days to weeks resulting in a delay in diagnosing and initiating treatment in infected individuals. The highly infectious nature of arthroconidia produced by *Coccidioides* species presents a safety risk to the laboratory personnel if a culture is not quickly identified and handled appropriately under biosafety level 3 containment. It is important to note that laboratory-acquired infections due to *Coccidioides* species have been reported in the literature [26]. A rapid *Coccidioides* DNA test could become an adjunct surveillance tool to study the incidence, prevalence, and overlapping features of Valley fever caused by *Coccidioides* species [7,8]. In summary, our modified diagnostic conventional PCR assay was excellent for the culture identification of *C. immitis* and *C. posadasii*, and the new duplex real-time PCR assay had broader utility for identifying two *Coccidioides* species from culture and primary specimens. The *Coccidioides* spp. duplex real-time PCR will allow rapid differentiation of *C. immitis* and *C. posadasii* from clinical specimens and further augment coccidioidomycosis treatment and surveillance.

## Supporting information

**S1 Fig. Multiple alignments of contigs and *PRA2* gene for the design of primers and probes for conventional and real-time PCR assays. (A)** Nucleotide sequences of PCR amplicons of *C. immitis* (Ci45815 & Ci45816) and *C. posadasii* (Cp45809 & Cp45810) contigs [17] were aligned using Geneious R9 9.1.6 software (Biomatters, Inc., San Diego, CA). The gray background defines the consensus sequences between *C. immitis* and *C. posadasii* while non-consensus sequences are highlighted in color. Conventional PCR forward (V2119-F) and reverse (V2020-R) primers were designed from the consensus region flanking the deleted region (86-bp) of *C. posadasii*. As a result, primer set produced 114-bp product for *C. posadasii* and 200-bp product for *C. immitis*. **(B)** *PRA2* gene from both *C. immitis* and C. *posadasii* were aligned and consensus region of the gene was used for the design of forward (V1753-F), and reverse (V1754-R) primers, and probe (V1755). The resulting primers and probe produced amplicons against both *C. immitis* and *C. posadasii*. **(C)** *C. immitis* (Ci45815 & Ci45816) and *C. posadasii* (Cp45809 & Cp45810) contigs were aligned as described in S1A Fig. The forward primer (V2116-F) was designed from *C. immits* contig region deleted in *C. posadasii* while reverse primer (V2118-R) and probe (V2117) were designed from consensus region of the contigs. The resulting primers and probe produced amplicon against *C. immits*, but not against *C. posadasii*.
(TIF)

**S1 Table. *Coccidioides* duplex real-time PCR assay specificity.** Genomic DNA from pathogenic fungi both closely and distantly related to *Coccidioides* species were tested in duplicate. A few bacterial pathogens, usually associated with common respiratory diseases, were also included in specificity panel. DNA from none of the fungal or bacterial isolates yielded

detectable fluorescence, and thus, confirming the high specificity of *Coccidioides* duplex real-time PCR assay.
(XLSX)

**S2 Table. Comparison of the diagnostic and *Coccidioides* duplex real-time PCR assay for detection of *Coccidioides* DNA from primary human and animal specimens.** DNA was extracted from both fixed and unfixed human and animal primary specimens and subjected to diagnostic PCR and *Coccidioides* duplex real-time PCR assay. Results revealed the duplex real-time PCR assay was superior to conventional diagnostic PCR for the detection of *C. immitis* or *C. posadasii* from various sample types investigated.
(XLSX)

**S3 Table. Assay reproducibility of *Coccidioides* duplex real-time PCR. (A) Inter-assay reproducibility:** DNA of varying concentrations from three isolates each of *C. posadasii* and *C. immitis* were tested in duplicate on 3 separate days. The Ct values obtained were consistent with % CV of <5%, which confirmed an excellent inter-assay reproducibility, **(B) Intra-assay reproducibility:** DNA samples from three isolates each of *C. immitis* and *C. posadasii* with varying concentrations were tested in triplicate on the same day on same plate in one run. The Ct values obtained were consistent with % CV of <5%, which confirmed an excellent intra-assay reproducibility.
(XLSX)

**S4 Table. Blinded panel of *Coccidioides* duplex real-time PCR.** The assay's precision was assessed using blinded panel with varying concentration of gDNA from 20 isolates each of *C. posadassii* and *C. immitis*, and 10 isolates of fungi other than *Coccidioides* spp. The duplex assay produced Ct values corresponding to input gDNA of either *C. immitis* or *C. posadasii* but not to other fungal species, confirming assay accuracy.
(XLSX)

**S5 Table. Diagnostic PCR and *Coccidioides* duplex real-time PCR testing of *Coccidioides* isolates from California.** Genomic DNA from 17 *C. immitis* isolates, received from California Department of Public Health, were evaluated by conventional diagnostic and duplex *Coccidioides* real-time PCR assay. Of 17 isolates, 13 were confirmed as *C. immitis* and four as *C. posadasii* by both assays, further confirming the utility of these assays for culture identification of both these species.
(XLSX)

**S6 Table. Evaluation of archived CSF and pleural fluid samples by *Coccidioides* duplex real time PCR assay.** DNA extracted from 15 residual CSF and pleural fluid samples, which were positive by serology tests (immunodiffusion & complement fixation), were tested by *Coccidioides* duplex real-time PCR assay. Of these samples, only two were positive for *C. posadasii* DNA by real-time PCR, suggesting a possible low correlation with the serology tests.
(XLSX)

## Acknowledgments

We acknowledge the Wadsworth Center (WC) Tissue Culture & Media, and the Applied Genomic Technologies Cores for media and sequencing services. Dr. Roberta Wallace, Milwaukee Zoo, is thanked for providing Rhino tissues suspected of coccidioidomycosis. Dr. Kimberlee McClive-Reed provided editorial comments on the draft manuscript. We thank YanChun Zhu for assistance with the reference testing of *Coccidioides* spp, and Lynn Leach for

helping with one set of data interpretations. VC thanks Dr. Edward Desmond and other staff members at the California Department of Public Health for assistance with implementation of *Coccidioides* real-time PCR test. The contents of this manuscript are solely the responsibility of the authors and do not necessarily represent the official views of the NYSDOH or the CDC.

## Author Contributions

**Conceptualization:** Sudha Chaturvedi, Vishnu Chaturvedi.

**Data curation:** Sudha Chaturvedi, Tanya R. Victor, Anuradha Marathe, Ketevan Sidamonidze.

**Formal analysis:** Tanya R. Victor, Anuradha Marathe, Ketevan Sidamonidze.

**Funding acquisition:** Sudha Chaturvedi, Vishnu Chaturvedi.

**Investigation:** Tanya R. Victor, Anuradha Marathe, Ketevan Sidamonidze.

**Methodology:** Tanya R. Victor, Anuradha Marathe, Ketevan Sidamonidze, Kelly L. Crucillo.

**Project administration:** Sudha Chaturvedi, Vishnu Chaturvedi.

**Resources:** Sudha Chaturvedi, Kelly L. Crucillo, Vishnu Chaturvedi.

**Software:** Tanya R. Victor, Anuradha Marathe, Ketevan Sidamonidze.

**Supervision:** Sudha Chaturvedi.

**Validation:** Tanya R. Victor, Anuradha Marathe, Ketevan Sidamonidze.

**Visualization:** Sudha Chaturvedi, Tanya R. Victor, Anuradha Marathe, Ketevan Sidamonidze, Kelly L. Crucillo, Vishnu Chaturvedi.

**Writing – original draft:** Sudha Chaturvedi.

**Writing – review & editing:** Sudha Chaturvedi, Tanya R. Victor, Vishnu Chaturvedi.

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
