## [Decision Letter · Decision Letter 0]

24 Jun 2021

Dear Dr. Chaturvedi,

Thank you very much for submitting your manuscript "Real-time PCR assay for detection and differentiation of Coccidioides immitis and Coccidioides posadasii from culture and clinical specimens" for consideration at PLOS Neglected Tropical Diseases. As with all papers reviewed by the journal, your manuscript was reviewed by members of the editorial board and by several independent reviewers. In light of the reviews (below this email), we would like to invite the resubmission of a significantly-revised version that takes into account the reviewers' comments. 

We cannot make any decision about publication until we have seen the revised manuscript and your response to the reviewers' comments. Your revised manuscript is also likely to be sent to reviewers for further evaluation.

Sincerely,

Angel Gonzalez, Ph.D.

Associate Editor

Fabiano Oliveira

Deputy Editor

Reviewer's Responses to Questions

**Key Review Criteria Required for Acceptance?**

**Methods**

-Are the objectives of the study clearly articulated with a clear testable hypothesis stated?

-Is the study design appropriate to address the stated objectives?

-Is the population clearly described and appropriate for the hypothesis being tested?

-Is the sample size sufficient to ensure adequate power to address the hypothesis being tested?

-Were correct statistical analysis used to support conclusions?

-Are there concerns about ethical or regulatory requirements being met?

Reviewer #1: The objective of the study are clearly articulated. But this study is no a hypothesis based study. It is a methods based study. The study design does address the stated objectives.

The samples and populations are clearly described but a hypothesis is not being tested.

The statistical test were not adequately described. On Page 10 in the Statistical analysis section, the authors should state the statistical test performed e.g., t-test, anova etc. including the post-hoc tests.

Yes, the ethical and regulatory requirements seem to be met.

Reviewer #2: While statistical testing was briefly mentioned, it is extremely vague and no specific tests were mentioned. No evidence from statistical tests are given in any of the tables.

Reviewer #3: Chuterverdi et al have provided a very nice article on the development of RT-PCR testing for Coccidioides spp. There methods are sound, all relevant information is provided, and their labs has a history of strong work in similar areas. I do not have any methodologic concerns.

**Results**

-Does the analysis presented match the analysis plan?

-Are the results clearly and completely presented?

-Are the figures (Tables, Images) of sufficient quality for clarity?

Reviewer #1: Yes.

Reviewer #2: (No Response)

Reviewer #3: Results are very easy to follow, are justified by the data, and clear.

**Conclusions**

-Are the conclusions supported by the data presented?

-Are the limitations of analysis clearly described?

-Do the authors discuss how these data can be helpful to advance our understanding of the topic under study?

-Is public health relevance addressed?

Reviewer #1: Yes.

Reviewer #2: (No Response)

Reviewer #3: Conclusions are sound. Dsicussion should be improved a bit with additional references added, and the clinical usefulness of RT-PCR discussed. The results of Barker et al with regard to spp specific differences and how this may be of clinical significance (not just epi significance) can be added.

**Editorial and Data Presentation Modifications?**

Reviewer #1: On Page 10 in the Statistical analysis section, the authors should state the statistical test performed e.g., t-test, anova etc. including the post-hoc tests.

Page 11: Lines 1-2 Amplification efficiency is important information to this study because the efficiency can impact the limit of detection. Amplification efficiency will also allow reviewers to access how reliable the assay is. Good quality assay typically have amplification efficiencies of 90 - 110 %. Please, add this data.

Reviewer #2: The methods can be difficult to follow at times. It would benefit from subheadings, especially in the DNA extraction section, to differentiate between protocols for the various sample types. The manufacturers' info is not always complete in this section and the formatting of units is not consistent.

The PCR sections may benefit from diagrams, particularly for the primer and probe design.

Many of the tables contain raw data, rather than summaries for sample types, and are not interpretation-friendly.

The manuscript would benefit if the raw data is kept in the supplement and interpretations presented in the main manuscript.

Reviewer #3: Italicize Coccidioides throughout (page 5 – line 3 etc; why were continuous line numbers not used throughout? Far simpler for reviewers).

Page 15, lines 3-8; this has previously been described by Taylor et al and Barker et al. Appropriate references could be added.

**Summary and General Comments**

Reviewer #1: What are the main claims of the paper and how significant are they for the discipline?

Coccidioidomycosis (Valley Fever) is a pulmonary and systemic fungal disease with increasing incidence and expanding endemic areas. The differentiation of two Coccidioides species remains problematic in clinical laboratories as conventional methods are limited by complexity or time constraints. Therefore, the authors developed TaqMan-probes for a duplex qPCR assay for rapid differentiation of C. immitis and C. posadasii from culture and clinical specimens.

Are these claims novel? If not, which published articles weaken the claims of originality of this one?

Although the techniques used here are not novel per se, the TaqMan-probes developed here overcome short comings of current techniques for diagnosis of Coccidioidomycosis.

Are the claims properly placed in the context of the previous literature? Have the authors treated the literature fairly?

The claims appear to be properly placed in the context of previous literature.

Do the data and analyses fully support the claims? If not, what other evidence is required?

The data partially support the claims made here. 

On Page 10 in the Statistical analysis section, the authors should state the statistical test performed e.g., t-test, anova etc. including the post-hoc tests. 

Would additional work improve the paper? How much better would the paper be if this work were performed and how difficult would it be to do this work?

I don’t believe that additional work is necessary.

PLOS Neglected Tropical Diseases encourages authors to publish detailed protocols and algorithms as supporting information online. Do any particular methods used in the manuscript warrant such treatment? If a protocol is already provided, for example for a randomized controlled trial, are there any important deviations from it? If so, have the authors explained adequately why the deviations occurred?

No.

Is this paper outstanding in its discipline? If yes, what makes it outstanding? If not, why not?

The authors have developed a technique with advantages over current methodologies.

If the paper is considered unsuitable for publication in its present form, does the study itself show sufficient potential that the authors should be encouraged to resubmit a revised version?

No. With very minor revision the paper is suitable for publication.

Are original data deposited in appropriate repositories and accession/version numbers provided for genes, proteins, mutants, diseases, etc.?

N/A

Are details of the methodology sufficient to allow the experiments to be reproduced?

Page 11: Lines 1-2 Amplification efficiency is important information to this study because the efficiency can impact the limit of detection. Please, add this data instead of stating data not shown.

Is the manuscript well organized and written clearly enough to be accessible to non-specialists?

Yes.

Reviewer #2: Financial disclosure section has not been completed to the specifications of the journal.

Reviewer #3: (No Response)

PLOS authors have the option to publish the peer review history of their article (what does this mean?). If published, this will include your full peer review and any attached files.

Reviewer #1: No

Reviewer #2: No

Reviewer #3: No
---

## [Editor Report · Decision Letter 1]

25 Aug 2021

Dear Dr. Chaturvedi,

We are pleased to inform you that your manuscript 'Real-time PCR assay for detection and differentiation of Coccidioides immitis and Coccidioides posadasii from culture and clinical specimens' has been provisionally accepted for publication in PLOS Neglected Tropical Diseases.

Best regards,

Angel Gonzalez, Ph.D.

Associate Editor

Fabiano Oliveira

Deputy Editor

---

## [Editor Report · Acceptance letter]

10 Sep 2021

Dear Dr. Chaturvedi,

We are delighted to inform you that your manuscript, "Real-time PCR assay for detection and differentiation of Coccidioides immitis and Coccidioides posadasii from culture and clinical specimens," has been formally accepted for publication in PLOS Neglected Tropical Diseases.

Best regards,

Shaden Kamhawi

co-Editor-in-Chief

Paul Brindley

co-Editor-in-Chief
